# Articulating Hip Spacers with a Constrained Acetabular Liner: Effect of Acetabular Bone Loss and Cementation Quality

**DOI:** 10.3390/antibiotics12091384

**Published:** 2023-08-30

**Authors:** Grayson T. Glenn, Andrew E. Apple, Simon C. Mears, C. Lowry Barnes, Benjamin M. Stronach, Eric R. Siegel, Jeffrey B. Stambough

**Affiliations:** 1Department of Orthopaedic Surgery, University of Arkansas for Medical Sciences, 4301 West Markham Street, Little Rock, AR 72205, USA; gtglenn@uams.edu (G.T.G.); scmears@uams.edu (S.C.M.); clbarnes@uams.edu (C.L.B.); bstronach@uams.edu (B.M.S.); 2Department of Orthopaedic Surgery, Tulane University, 1430 Tulane Avenue, New Orleans, LA 70112, USA; aapple36@gmail.com; 3Department of Biostatistics, University of Arkansas for Medical Sciences, 4301 West Markham Street, Little Rock, AR 72205, USA; siegelericr@uams.edu

**Keywords:** articulating antibiotic hip spacer, periprosthetic joint infection, cemented constrained liner, total hip arthroplasty, revision hip arthroplasty

## Abstract

Articulating hip spacers for periprosthetic joint infection (PJI) offer numerous advantages over static spacers such as improved patient mobilization, hip functionality, and soft tissue tension. Our study aimed to determine complication rates of a functional articulating spacer using a constrained liner to determine the role of acetabular cementation mantle and bone loss on the need for second-stage surgery. A retrospective review of 103 patients was performed and demographic information, spacer components and longevity, spacer-related complications, reinfection rates, and grade of bone loss and acetabular cement mantle quality were determined. There was no significant difference in spacer-related complications or reinfection rate between PJI and native hip infections. 33 of 103 patients (32.0%) elected to retain their spacers. Between patients who retained their initial spacer and those who underwent reimplantation surgery, there was not a significant difference in cement mantle grade (*p* = 0.52) or degree of bone loss (*p* = 0.78). Functional articulating antibiotic spacers with cemented constrained acetabular liners demonstrate promising early results in the treatment of periprosthetic and native hip infections. The rate of dislocation events was low. Further efforts to improve cement fixation may help decrease the need for second-stage reimplantation surgery.

## 1. Introduction

Periprosthetic joint infection (PJI) is a feared complication in total hip arthroplasty (THA), with an infection rate of approximately 1–2% in patients who have undergone THA surgery [1,2,3,4]. Infection can damage bone stock and affect underlying bone integrity, thus limiting the potential success of future revision procedures [5,6,7,8]. The gold standard treatment for THA PJI is two-stage revision arthroplasty with implantation of an antibiotic spacer [9,10,11,12,13]. More recently, implants have been utilized in an attempt to provide a functional spacer that may be able to last for a longer period of time or potentially be permanent, without the need for a second surgery [14]. This approach uses cemented hip implants to allow for immediate weight-bearing and function [5,9,15,16]. However, functional spacers are subject to their own subset of complications, such as prosthetic dislocation and fracture [9,11,17,18,19,20].

Dislocation rates as high as 41% have been reported for articulating spacer constructs [21]. Factors such as poor soft tissue quality and trochanteric insufficiency may exist in patients undergoing surgery that increase the instability of the construct and risk of dislocation. One alternative to reduce dislocation rates in such patient populations is the use of a constrained acetabular liner [22]. Constrained liners may put more stress on the fixation of the acetabular component. Articulating spacers are placed using the cement technique and controversy exists over the methodology of cementation. Better cementation may provide a longer-lasting implant but makes removal of cement more difficult if a second-stage surgery is required.

The purpose of our study was to describe our results utilizing a cemented metal hip stem with an all-polyethylene constrained tripolar liner, secured with surgeon-mixed, high-dose antibiotic cement in the staged treatment of both PJI and native hip infections. Comparisons between PJI and native hip infection groups were made as patients with both types of infections are candidates for antibiotic spacer placement. We intended to examine rates of second surgery and dislocation rates of the construct. In addition, we examined the role of acetabular cement mantles and acetabular bone loss to predict the longer survival of the spacer. We hypothesized that a higher-grade acetabular cement mantle and a lesser degree of acetabular bone loss would be predictors of longer-term spacer survival. Our rationale was that due to improved fixation of components with high-grade cementation and superior scaffolding provided by healthier acetabular bone stock, a more robust implant interface would be created to prevent complications such as dislocation and fracture [7,8,14].

## 2. Methods

After institutional review board approval, a retrospective review was conducted to identify patients who underwent placement of a functional, articulating hip spacer with a cemented, constrained, all-polyethylene acetabular liner and a metal femoral stem. These components were used in the treatment of PJI or native hip infection at a single tertiary care academic medical center from January 2016 through April 2020. Inclusion criteria were the placement of the articulating spacer and age greater than 18 years. Patients were excluded if no follow-up information was available in the health record (6) or if additional hardware such as a cage was implemented to provide stability to the implant (4). Our final cohort consisted of 103 articulating hip spacers placed from January 2016 to April 2020. Seventy-nine spacers were placed for PJI while 24 were placed for native hip chronic septic arthritis.

All infected hips underwent prosthesis removal and/or resection of infected bone and thorough irrigation and debridement of the hip. All hips were irrigated with at least 6 L of saline and a betadine solution using pulse lavage. A cemented, constrained all-polyethylene acetabular liner was used (Trident constrained liner, Stryker, Mahwah, NJ, USA). A cemented femoral implant and metal head were used (OmniFit, Stryker, Mahwah, NJ, USA). Both components were secured with 2–3 batches of plain Cobalt^®^ bone cement (DJO Surgical, Austin, TX, USA) with surgeon-directed manual addition of 2–3 g of vancomycin and 2.4–3.6 g of tobramycin per batch of cement. On the acetabular side, acetabular reamers were used and the component was undersized by either 4 or 6 mm to gain a proper cement mantle. Femoral components were cemented only in the metaphyseal region to aid in future removal at the time of second-stage reimplantation. Cement restrictors were not used and a hand-packing technique was performed. Broaches were utilized to determine stem size and a size one under the broach was selected. Cobalt chrome femoral heads were used in all cases and the size was used in concordance with the acetabular liner. Following spacer placement, all patients were treated with 6 weeks of culture-directed intravenous antibiotics based on recommendations from our Musculoskeletal Infectious Disease Service. Post-operative weight-bearing status was determined per the surgeon’s discretion. Second-stage surgery was at the discretion of the operating surgeon. Surgeries were performed by 4 arthroplasty surgeons who have different philosophies of treatment. One surgeon was more likely to recommend early revision and the others performed a second surgery only if patients were symptomatic.

Demographic information is recorded in Table 1. Microbiology cultures were positive in 79 of 103 patients and organisms are shown in Table 2. Spacer components, spacer-related complications such as component failure, prosthetic dislocation, and periprosthetic fracture, rates of recurrent infection, and spacer longevity were recorded. Grading of acetabular bone loss was performed according to Paprosky classifications [23] and cement mantle quality was measured using postoperative anteroposterior X-rays according to the criteria proposed by Hodgkinson, Shelley, and Wroblewski [24]. A radiographic example of a prosthesis with a high-grade cement mantle and no acetabular bone loss can be seen in Figure 1, contrasted with a low-grade cement mantle in the setting of moderate acetabular bone loss in Figure 2. Spacer longevity was measured for patients who underwent second-stage reimplantation and for those with a retained spacer. Comparisons were also made between spacers placed for PJI and native hip septic arthritis. Within the patient subset who received second-stage reimplantation, factors leading to reimplantation such as component loosening and pain were analyzed. Rates of reinfection and spacer-related complications were compared between patients who retained their initial spacer and those who underwent a second-stage reimplantation, in addition to grading of bone loss and cement mantle quality between the two groups.

SAS v.9.4 software (SAS Institute Inc., Cary, NC, USA) was deployed for statistical analysis. Age and BMI were compared for group differences with the Kruskal–Wallis test, while all other variables were compared for group differences with Fisher’s exact test. *p* < 0.05 was considered statistically significant.

## 3. Results

At an average follow-up of 466.9 days, 70/103 (68.0%) of hips had undergone second-stage reimplantation (mean 151.9 days), while 33/103 (32.0%) had retained the spacer (mean duration 257.0 days). The majority of hips (94/103, 91.3%) were free from reinfection at one year with no statistically significant difference between the PJI (92.4%) and native groups (87.5%, *p* = 0.43).

There were 10 (9.7%) spacer-related complications, with no significant difference in spacer-related complication rate between PJI and native infections (7/79 PJI vs. 3/24 native, *p* = 1.0). Spacer-related complications included acetabular component dislodgement (3), spacer dislocation (3), prosthetic dislocation (1), periprosthetic fracture (1), hematoma requiring reoperation (1), and femoral component loosening (1) (Table 3). Spacer-related complications were observed in 9/72 (13%) of patients who received second-stage reimplantation, as opposed to 1/31 (3%) in those who retained their spacer (*p* = 0.275).

A slightly greater number of patients who chose to retain their articulating constrained spacer had a cement mantle grade of 0 or 1 (15/31, 48%) as opposed to patients who underwent second-stage reimplantation (29/72, 40%) (Table 4). Most patients with retained spacers had a cement mantle grade of 2 or greater (16/31, 52%) as well as patients who underwent second-stage reimplantation (43/72, 60%, Table 4).

No acetabular bone loss (20) or mild loss (8) was observed in 28/31 (90%) of patients with retained spacers, compared to 59/72 (82%) of patients who underwent second-stage reimplantation (Table 5). In the retained spacer group, 2/31 (6%) of patients had moderate or severe acetabular bone loss, compared to 10/72 (14%) of patients who underwent second-stage reimplantation (Table 5).

## 4. Discussion

Periprosthetic hip joint infection represents a significant health burden to patients who have undergone total hip arthroplasty. In treating PJI, articulating antibiotic spacers have become popular for their numerous advantages for patient functionality. However, articulating spacers are subject to their own subset of complications, with fracture and periprosthetic dislocation being especially problematic. Earlier studies have shown fracture rates of 13.6% [19] and rates of dislocation as high as 41% [21]. By utilizing an articulating construct with a cemented constrained polyethylene acetabular liner and cemented femoral stem, we were able to minimize the risk of dislocation. The rate in our series was 5.8%. This is compared to 26% in the literature [25]. The use of a constrained liner is thought to put additional stress on the bone–implant interface [26]. Rates of acetabular implant loosening were 3.9%.

Recurrent infection was seen in both PJI (7.6%, 6/79) and native (12.5%, 3/24) infection groups. The literature has shown a reinfection rate of approximately 10% with two-stage treatment of PJI [27]. Rates after native infections with arthritic change are less understood, but in a study by Fleck et al. were shown to be 7.2% [28]. Persistent infection of the hip joint after surgical treatment is associated with significant morbidity [29,30], often leading to patients undergoing repeat two-stage procedures which have been associated with low success rates [31].

Our implant retention at one year was only 32.0%. This may be somewhat due to differences in surgeon techniques or surgical decision-making. One of the treating surgeons plans reimplantation at 3 months while the other three surgeons are more likely to allow patients to keep their spacers if they are functioning well clinically. Tsung et al. described the 1.5-stage revision wherein 44.7% of patients kept a spacer that was functional [32]. There may be a higher degradation level of retention as time goes on with additional loosening of implants. A case series by Choi et al. described long-term outcomes of retained articulating spacers fixated with antibiotic-laden cement, with 83.3% (15/18) of patients maintaining articulating spacers for up to 6 years with adequate functionality [33]. Additionally, a case report by Luk et al. presented a patient who has retained an articulating spacer for 6 years with no evidence of infection, loosening of the spacer construct, or periprosthetic fracture [34]. Another possible solution is a one-stage revision using uncemented revision implants or a better cement technique.

The quality of the cement technique did not seem to relate to reimplantation rates. There was no significant difference in cement mantle grade between those who kept their prosthesis and those who underwent reimplantation surgery. Improvements in technique may help this; however, at the expense of more difficulty later if a revision is required. Bone loss in the acetabulum was not found to be a risk factor for revision but may make revision surgery more difficult.

In addition to advantages in patient functionality, articulating spacers may provide an effective longer-term solution for patients who are unfit to undergo reimplantation surgery or who elect to retain their spacer, allowing patients a satisfactory quality of life while preventing the physical and psychological burden of repeat operation [35]. This is an important consideration, especially in frail patients [36]. The articulating spacer construct appears to be a feasible long-term solution as a significant number of patients elected to retain their initial spacer. Within the patient subset who retained their spacers, cement mantles were of a slightly higher grade, with 48% of patients having either a grade 0 or grade 1 cement mantle and a lesser degree of acetabular bone loss was observed when compared with patients who underwent eventual second-stage reimplantation, with 90% of patients having only mild bone loss or none at all. This suggests that the spacer construct is reliable as a long-term solution given cement mantles and bone stock are adequate at the time of implantation.

The burden of a second-stage revision procedure on patients is an important consideration. In attempting to solve this problem, a 1.5-stage revision surgery has been adopted by some centers for the treatment of infected total knee arthroplasty, allowing for patients to have a functional articulating spacer construct placed permanently with comparable infection eradication rates to those seen in retained articulating spacers [37,38]. Additionally, this method offers greater ease of resection and preservation of bone should a patient need a second surgery due to reinfection, construct loosening, or functional decline [38]. In the future, having a similar construct available for the treatment of hip PJI would be advantageous if rates of infection eradication and construct longevity were comparable to articulating constructs in current use.

Our study has some limitations. The use of our articulating spacer is off-label and requires intraoperative surgeon adjustments depending on bone loss and cement supplementation needs. Because our surgeons intentionally do not cement the entire femur, we were unable to completely assess cement mantle quality and the subsequent need for revision surgery. Our study is a retrospective review of data. Surgeons had different philosophies regarding long-term spacer usage. The decision to reimplant versus leave in situ is often a complex, shared-decision process that considers many patient and implant factors.

## 5. Conclusions

Articulating antibiotic spacers with cemented constrained acetabular liners and metal femoral components represent a potential option for some patients in the treatment of PJI and native hip joint infections. The articulating spacer construct is able to eradicate hip joint infection while preserving good patient functional status. Articulating antibiotic spacers may be a feasible long-term solution, especially in frail patients. This spacer construct appears to be reliable with regard to component dislocation. The quality of acetabular cement mantles and acetabular bone loss were not related to the need for a second surgery.

## Figures and Tables

**Figure 1 antibiotics-12-01384-f001:**
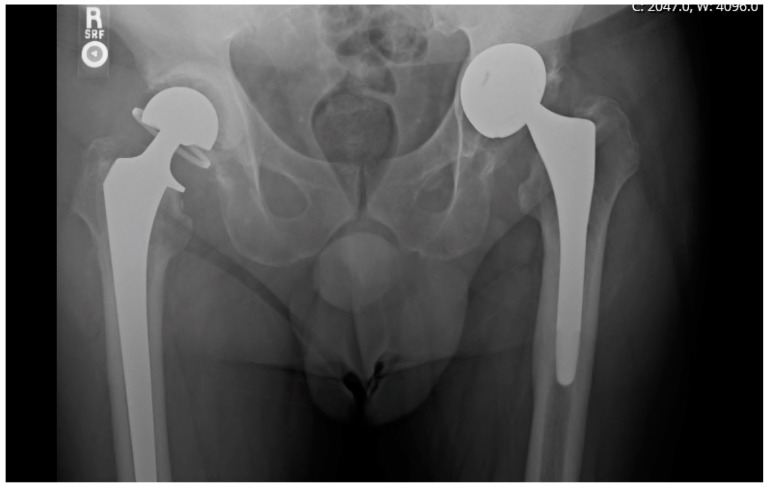
Articulating spacer construct with a high-grade cement mantle and no acetabular bone loss in a patient who elected to retain their initial spacer.

**Figure 2 antibiotics-12-01384-f002:**
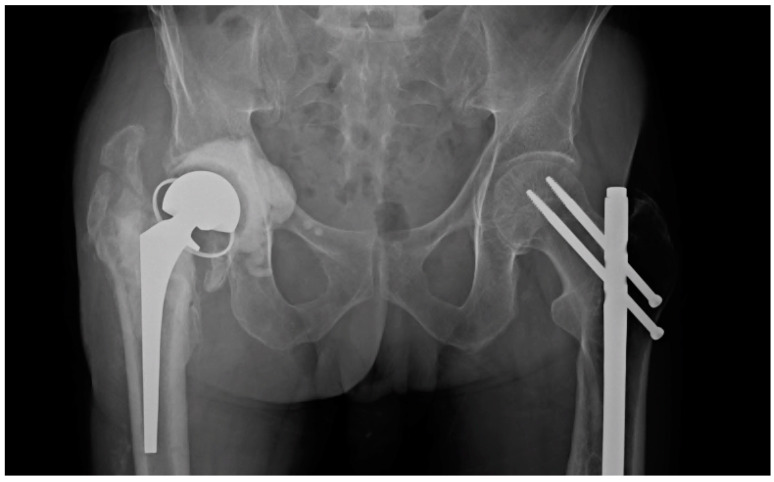
Articulating spacer construct in a patient with a low-grade cement mantle and moderate acetabular bone loss.

**Table 1 antibiotics-12-01384-t001:** Demographic factors for patients.

	All Spacers (n = 103)	PJI (n = 79)	Native (n = 24)	*p*-Value
Age	62.1 (SD 13.2)	63.3 (SD 13.3)	58.5 (SD 12.7)	0.124
Sex				
Male	51 (49.5%)	41 (51.9%)	10 (41.7%)	0.486
Female	52 (50.5%)	38 (48.1%)	14 (58.3%)	
BMI	31.6 (SD 8.9)	32.5 (SD 9.3)	28.8 (SD 6.8)	0.088
Side				
Right	46 (44.7%)	40 (50.6%)	6 (25.0%)	0.035
Left	57 (55.3%)	39 (49.4%)	18 (75.0%)	

**Table 2 antibiotics-12-01384-t002:** Culture results of the cases.

Organism	Total Number of Infections	PJI	Native
MSSA	26	23	3
MRSA	18	13	5
Staphylococcus epidermidis	9	8	1
MRSE	5	4	1
Candida albicans	4	3	1
Klebsiella pneumoniae	3	3	0
Enterococcus faecalis	4	3	1
Escherichia coli	2	1	1
Pseudomonas aeruginosa	2	2	0
Streptococcus anginosus	3	3	0
Staphylococcus lugdunensis	3	1	2
Enterobacter cloacae	2	2	0
Acinetobacter baumanii	1	1	0
Candida parapsilosis	1	1	0
Cladosporium	1	1	0
Corynebacterium	1	1	0
Granulicatella adiacens	1	1	0
Enterobacter aerogenes	1	1	0
Klebsiella oxytoca	1	1	0
Leuconostoc mesenteroides	1	0	1
Morganella morganii	1	1	0
Proteus mirabilis	1	1	0
Streptococcus agalactiae	1	1	0

**Table 3 antibiotics-12-01384-t003:** Comparison of spacer-related complications and rates of recurrent infection in PJI and native hip infection groups.

	All Spacers (n = 103)	PJI (n = 79)	Native (n = 24)
Spacer-related complications	10 (9.7%)	7 (8.9%)	3 (12.5%)
Acetabular component dislodgement	3	2	1
Spacer dislocation	3	1	2
Prosthetic dislocation	1	1	0
Periprosthetic fracture	1	1	0
Postop hematoma/bleeding requiring operation	1	1	0
Femoral component loosening	1	1	0

**Table 4 antibiotics-12-01384-t004:** Comparison of cement mantle grades between patients who underwent second-stage reimplantation and those who retained their spacer.

Cement Mantle Grade	All Spacers	Reimplant	Retained	*p*-Value
0 or 1	44 (43%)	29 (40%)	15 (48%)	0.517
2 or greater	59 (57%)	43 (60%)	16 (52%)	

**Table 5 antibiotics-12-01384-t005:** Comparison of acetabular bone loss as defined by the Paprosky classification between patients who underwent second-stage reimplantation and those who retained their spacer.

Acetabular Bone Loss	All Spacers	Reimplant	Retained	*p*-Value
None	65 (63%)	45 (63%)	20 (65%)	0.781
Mild	22 (21%)	14 (19%)	8 (26%)	
Moderate	9 (9%)	7 (10%)	2 (7%)	
Severe	3 (3%)	3 (4%)	0 (0%)	
Preexisting Mild	2 (2%)	1 (1%)	1 (3%)	
Preexisting Moderate	2 (2%)	2 (3%)	0 (0%)	
Preexisting Severe	0 (0%)	0 (0%)	0 (0%)	

## Data Availability

The data is stored securely on our encrypted BOX platform per our IRB designation.

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
