# Peer review of "Articulating Hip Spacers with a Constrained Acetabular Liner: Effect of Acetabular Bone Loss and Cementation Quality"

_antibiotics, 2023, doi:10.3390/antibiotics12091384_

Round 1

Reviewer 1 Report

The study evaluates the complication rates of an articulating hip spacer using a constrained liner. Complications include component failure, prosthetic dislocation, periprosthetic fracture, rates of recurrent infection, and spacer longevity were examined. Comparisons were made between spacers for periprosthetic joint infections and native hip infections. The study showed some interesting results. However, the following concerns need to be addressed:

  1. In the abstract, please introduce the full name of PJI the first time mentioned it.
  2.  In the introduction, the authors should mention the significance of comparing the two. Why did the authors compare PJI and native hip infections?
  3. Why the authors compare PJI and native hip infections in Table 2 and 3; however, the two are not compared for cement mantle grade and acetabular bone loss study(Table 4-5).
  4. The authors mentioned, "Surgeries were performed by 4 arthroplasty surgeons who did have different philosophies of treatment ". The author may need to consider this factor and see if the conclusion still holds for each surgeon. 
  5. In demographic factors for patients in Table 1, the Left side and the Right side are significantly different. How would this factor impact the results?

Author Response

Reviewer #1

In the abstract, please introduce the full name of PJI the first time mentioned it.

    1. See abstract. Full name of PJI has now been provided in the first sentence.

  1.  In the introduction, the authors should mention the significance of comparing the two. Why did the authors compare PJI and native hip infections?
    1. These were compared due to the convenience of the sample. Comparison to static spacer would not have been reasonable due to lack of concern for cement mantle quality in static spacers and differences in forces that articulating spacers are subject to vs. static spacers. To compare subgroups within a set of patients who are candidates for articulating spacer placement made the most logical sense given similarities between the subgroups. See sentence added to introduction.
  2. Why the authors compare PJI and native hip infections in Table 2 and 3; however, the two are not compared for cement mantle grade and acetabular bone loss study(Table 4-5).
    1. These were compared due to our hypothesis that cement mantle quality and degree of bone loss would predict spacer survival, and analysis of reimplanted vs. retained spacer numbers was the best way to determine this, regardless of whether patients within these two groups also fell into PJI or native groups.
  3. The authors mentioned, "Surgeries were performed by 4 arthroplasty surgeons who did have different philosophies of treatment ". The author may need to consider this factor and see if the conclusion still holds for each surgeon. 
    1. Surgeons who performed the surgeries analyzed in this study were aware of and involved in the making of the study and were in agreement on conclusions stated within this manuscript.
  4. In demographic factors for patients in Table 1, the Left side and the Right side are significantly different. How would this factor impact the results?
    1. Prevalence of Left vs Right side is purely a function of our patient sample, and not representative of a trend in PJI or antibiotic spacer implantation. We do not anticipate that this had any appreciable effect on the results of our study. 

Reviewer 2 Report

Authors may consider representing the data graphically where ever possible.

A grammar check is advised

Author Response

Reviewer #2:

  1. Authors may consider representing the data graphically wherever possible.
    1. Authors of this paper felt that representation of data through tables was adequate for the data categories involved in this manuscript.
  2. A grammar check is advised.
    1. The paper was checked for grammar and any necessary edits were made.

Reviewer 3 Report

1. Some reasons and references for the hypothesis that "a better acetabular cement mantle and less bone loss would be predictors of longer-term spacer survival" should be included.

2. Please comment on the acetabular bone loss and cement mantle quality per se from the study in the conclusion.  This will enable the title to be in tune with the manuscript content.

3. Reference [24] does not refer to Hodgkinsons's classification.  Kindly check.

4. Fig. 1 and Fig. 2  are not mentioned in the manuscript.

Author Response

Reviewer #3:

  1. Some reasons and references for the hypothesis that "a better acetabular cement mantle and less bone loss would be predictors of longer-term spacer survival" should be included.
    1. See end of introduction section. Reasons and references added.
  2. Please comment on the acetabular bone loss and cement mantle quality per se from the study in the conclusion.  This will enable the title to be in tune with the manuscript content.
    1. Statement on acetabular bone loss and cement mantle quality was edited to be more clear. See final sentence of conclusion.
  3. Reference [24] does not refer to Hodgkinsons's classification.  Kindly check.
    1. Thank you for bringing this to my attention. The reference has been replaced, and the reference order has been adjusted accordingly.
  4. 1 and Fig. 2  are not mentioned in the manuscript.

See edit in Methods section. Mention of Figures 1 and 2 has been added.

Round 2

Reviewer 1 Report

The authors have addressed my comments.